# Simultaneous Detection of Five Foodborne Pathogens Using a Mini Automatic Nucleic Acid Extractor Combined with Recombinase Polymerase Amplification and Lateral Flow Immunoassay

**DOI:** 10.3390/microorganisms10071352

**Published:** 2022-07-05

**Authors:** Bei Jin, Biao Ma, Jiali Li, Yi Hong, Mingzhou Zhang

**Affiliations:** Zhejiang Provincial Key Laboratory of Biometrology and Inspection & Quarantine, China Jiliang University, Hangzhou 310018, China; yieio0219@163.com (B.J.); 16a0701109@cjlu.edu.cn (B.M.); qjc1993@126.com (J.L.); 18768152453@163.com (Y.H.)

**Keywords:** recombinant polymerase amplification, lateral flow immunoassay, multiple detection, foodborne pathogens, automatic extractor

## Abstract

In recent years, foodborne disease outbreaks have caused huge losses to the economy and have had severe impacts on public health. The accuracy and variety of detection techniques is crucial to controlling the outbreak and spread of foodborne diseases. The need for instruments increases the difficulty of field detection, while manually-handled samples are subject to user error and subjective interpretation. Here, we use a mini automatic nucleic acid extractor combined with recombinant polymerase amplification (RPA) and lateral flow immunoassay (LFIA) for simultaneous quantitative detection of five major foodborne pathogens. The pre-treatment device using the magnetic bead method allows for nucleic acid extraction of the reagent tank without manual operation, which is highly efficient and stable for preventing aerosol contamination. The *nuc* gene of *Staphylococcus aureus*, the *toxR* gene of *Vibrio parahaemolyticus*, the *rfbE* gene of *Escherichia coli O157:H7*, the *hlyA* gene of *Listeria monocytogenes,* and the *fimY* gene of *Salmonella enterica* were used as target fragments. The labeled antibody concentration is optimized on the LFIA to find the equilibrium point for the binding capacity of the five chemical markers and to efficiently and accurately visualize the bands. The RPA assay shows an optimal performance at 37 °C for 15 min. The optimized RPA-LFIA detection limit can reach 10^1^ CFU/mL. There was no cross-reactivity among forty-eight strains. Furthermore, the average recoveries in spiked food samples were 90.5–104.5%. In summary, the RPA-LFIA established in this study can detect five pathogenic bacteria simultaneously with little dependence on laboratory equipment, and it has promising prospects for screening in low-resource areas.

## 1. Introduction

With the rapid growth of global economics, food supply chains have become increasingly long. The cooperation among food-producing enterprises has become diversified and complicated, increasing the risk of food safety risks [1]. The occurrence of foodborne disease events has resulted in 2.2 million deaths [2]. Foodborne pathogens contain viruses, bacteria, parasites, fungi, and other substances [3]. According to a 2019 European Union (EU) report [4], bacteria were the main pathogenic factor in foodborne pathogens, accounting for 19% of them. In China, the most commonly diagnosed cause was microbial pathogens, which reached 41.7% [5]. Among them, *Vibrio parahaemolyticus* (*V. parahaemolyticus*), *Staphylococcus aureus* (*S. aureus*), *Salmonella enterica* (*S. enterica*), *Escherichia coli O157:H7* (*E. coli O157:H7*), and *Listeria monocytogenes* (*L. monocytogenes*) were the top five pathogenic microorganisms [1]. These five foodborne pathogenic bacteria are the main pathogenic bacteria found in aquatic products, meat products, milk, and other foods. The ingestion of the contaminated food results in severe diarrhea and dehydration, and even causes major risks such as meningitis, febrile gastroenteritis, and systemic infections in humans [6,7,8,9,10,11,12,13]. Therefore, early identification, monitoring and warning of foodborne pathogens are essential to prevent the occurrence of foodborne diseases. These initiatives could help to identify changing trends in specific diseases and reduce the risk factors and disease burden [14].

Traditional techniques for foodborne pathogens mainly include microbiologic analysis and immunological assay [15]. The former usually requires a combination of morphological observation and serological examination, which is time-consuming. The other methodology depends on the specific antibody and skilled operation, making it expensive and dependent on the availability of professionals [16]. With the development of molecular biology, nucleic acid amplification has emerged. In the past few decades, polymerase chain reaction (PCR) has been successfully applied in the testing of foodborne pathogens [17]. However, it requires fairly operational skills, precise temperature control, expensive equipment, and long pretreatment, making it difficult to promote in resource-limited settings [18]. In view of simplicity and rapidity, isothermal amplification technology might be the answer for monitoring with out-of-laboratory tests, such as loop-mediated isothermal amplification (LAMP) and recombinase polymerase amplification (RPA). Neither of these tests depends on sophisticated instruments, and they can both be conducted by people with simple training [19]. Although the LAMP assay can typically amplify the target gene within an hour at 65 °C, a difficulty with it is the false-positive results caused by aerosol pollution [20,21]. The RPA assay has been demonstrated to be a suitable method to amplify target DNA/RNA molecules [22]. RPA testing shows very few issues when detecting samples with many impurities. Compared with other isothermal methods, it requires lower incubation temperature and is therefore low in energy consumption. Another advantage is its shorter reaction time. During the amplification, the recombinase bound binds to the single-stranded primer to form a stable nucleoprotein filament, which is used to recognize the homologous sites of the template sequence and the primer sequence and open the double-stranded DNA [23]. The DNA target is then amplified by a strand displacement enzyme, resulting in exponential amplification. The simple process was performed at a constant temperature (37–42 °C) within 30 min. The RPA assay has efficiency, great specificity, faster speed, and reliable results, which is facilitated in the field more than in LAMP [24].

The amplification products of RPA are represented by gel electrophoresis, fluorescent signal monitoring and lateral flow immunoassay (LFIA). There are still some key issues to be considered in detection, including the multiplex, cost, training, and accuracy. In order to meet the requirements of on-site detection, the LFIA method was a reasonable choice. Gold nanoparticles (AuNPs) have been widely used in various fields of food safety in recent years [25]. To facilitate observation, LFIA-labeled colloidal gold combined with RPA has been developed for the detection of foodborne pathogens [24]. The analyte moved under capillary force, bound to AuNP-labeled receptors, and then it was captured by specific antibodies that were fixed at the detection line. It was easy to determine whether the target gene was present by observing the color of the T-line. By fixing different antibodies to identify the corresponding pending samples, multiple detection could be realized in one strip. Previously, no more than three targets were selected to be tested in the same experiment [26,27]. Meanwhile, the mini automatic nucleic acid extractor (Auto-Pure Mini, Allsheng Instruments Co. Ltd., Hangzhou, China) was used to extract the nucleic acids using magnetic particles with high efficiency and large throughputs, which are small and portable. In addition, the scanning results could be quantified by the test strip reader (TSR-200, Allsheng Instruments Co. Ltd., Hangzhou, China).

In this study, we developed a rapid, multiple, convenient, and sensitive RPA-LFIA method for the simultaneous detection of *V. parahaemolyticus*, *S. aureus*, *S. enterica*, and *E. coli O157:H7*, and *L. monocytogenes* by using a mini automatic nucleic acid extractor. The primer ratios, incubation temperature, reaction time, concentration of magnesium ions in the buffer, and concentration of antibodies on the strip were optimized to increase the detection efficiency. After the analytical sensitivity and specificity were determined, the RPA-LFIA assay was applied in actual food samples. The results showed that the proposed method could detect five bacterial pathogens simultaneously, which was suitable in field detection and provided more convenience for healthcare. Moreover, it generated a promising technological potential for the quantitative diagnostics of foodborne pathogens.

## 2. Materials and Methods

### 2.1. Bacterial Culture Preparation and DNA Extraction

Forty-eight strains were used in this study (Table 1), including *V. parahaemolyticus*, *S. aureus*, *S. enterica*, *E. coli O157:H7*, *L. monocytogenes*, and other foodborne strains. *S. enterica* and *S. aureus* strains were grown in Luria-Bertani broth (LB, Sangon, Shanghai, China). *V. parahaemolyticus* was cultured in alkaline peptone water (APW, Hopebio, Qingdao, China) supplemented with 3% NaCl. *L. monocytogenes* strains were grown on buffered *Listeria* enrichment broth (BLEB, ThermoFisher, Waltham, MA, USA). *E. coli O157:H7* was cultured in lauryl tryptose broth (LTB, ThermoFisher, Waltham, MA, USA). The others were cultured in LB broth. All strains were incubated at 37 °C for 20–24 h, and the colony-forming units (CFU) of corresponding plates were calculated. Bacterial concentration was determined by plate colony counting.

To simplify the processing, the mini automatic nucleic acid extractor (Auto-Pure Mini) was used as the sample pretreatment device. Its working mechanism (Figure 1a) was specifically adsorbing the target extracts by magnetic beads with modified chemical groups. The liquid sample (5 mL) or the tissue homogenates (20 mg) were added to 600 μL of lysate and then incubated for 30 min. After these steps, it was centrifuged at 12,000 rpm for 5 min. The supernatant was added to a 96-deepwell plate. Another 30 μL of the magnetic bead suspension was then added. After repeatedly and rapidly mixing, the nucleic acids were combined for 10 min, and the magnetic absorption time was 90 s. The magnetic beads, adsorbed with genomic DNA, were transferred into a 600 μL washing solution using a magnetic rod. After it was transferred to the 600 μL washing solution, and a lifting movement was performed for 120 s, followed by magnetic absorption for 90 s. This step was repeated twice. Finally, the magnetic rod was transferred to a 100 μL eluent for 10 min. After that, the purified nucleic acids were obtained. This technique has the advantages of fast extraction speed, stable results, and intelligent operation of the automatic instrument. The mobile phone app allows for remote editing and transmission of the instrument’s program and real-time view of the running log. The obtained DNA was subsequently quantified on a spectrophotometer (DU730, Beckman Coulter, Burea, CA, USA) and stored at −20 °C until use.

### 2.2. RPA Primers Design

The primers were designed based on the *toxR* gene (Genebank accession number: GQ228073.1) of *V. parahaemolyticus*, the *nuc* gene (Genebank accession number: EF529607.1) of *S. aureus*, the *fimY* gene (Genebank accession number: JQ665438.1) of *S. enterica*, the *rfbE* gene (Genebank accession: AE005429) of *E. coli O157:H7*, and the *hlyA* gene (GenBank accession: HM58959) of *L. monocytogenes.* There were no homologous sequences among these five target genes after the analysis by MegAlign software (LaserGene, DNASTAR Inc., Madison, WI, USA). Five sets of specific RPA primers were then designed using Primer Premier 5.0 software (Premier Biosoft, San Francisco, CA, USA) according to the TwistDx instruction manual. To distinguish different amplicons, the primers with target-specific labels were tagged with tetrachlorofluorescein (TET) and digoxin (primers for the *toR* gene), carboxy fluorescein (FAM) and digoxin (primers for the *nuc* gene), carboxytetramethylrhodamine (TAMRA) and digoxin (primers for the *fimY* gene), biotin and digoxin (primers for the *refbE* gene), and cyanine 5 (Cy5) and digoxin (primers for the *hlyA* gene). All primers (Table 2) were synthesized by Invitrogen Biotechnology Co. Ltd. (Shanghai, China).

### 2.3. Multiple RPA Procedure

RPA amplification was performed using a TwistAmp Basic kit (TwistDx, Cambridge, UK) according to the manufacturer’s instructions. All of the reaction mixtures had a final volume of 50 μL, which contained 25 μL 2× reaction buffer, 10 μL nuclease-free water, 2 μL of each of the specific primers (10 μM) for five bacterial pathogens, and 0.5 μL of each template. The mixture solution was added to a lyophilized enzyme pellet. With the addition of 2.5 µL magnesium acetate, the amplification reaction started after a brief vortex and centrifugation procedure. The mixed solution was incubated at 37 °C for 15 min and then placed on ice. Negative controls were treated with sterile water instead of the DNA template. The incubated RPA products were diluted 50 times using a running buffer, which contained PBS and 3% Tween 20. The strips were scanned using a strip reader and the values on the test lines were read.

### 2.4. Preparation of AuNPs and Lateral Flow Dipsticks Immunoassay

The AuNPs were prepared using sodium citrate tannin reduction. The particle size of AuNPs was controlled by adjusting the content of trisodium citrate [28]. The strip was connected by five parts with continuous superposition: sample pad, conjugate pad, nitrocellulose filter (NC) membrane, adsorption pad, and backing card. The AuNPs-anti-digoxigenin monoclonal antibody was sprayed onto conjugate pads. The five test lines (T-lines) used anti-TET monoclonal antibody (for detection of *V. parahaemolyticus* in T1), anti-FAM monoclonal antibody (for detection of *S. aureus* in T2), anti-TAMRA monoclonal antibody (for detection of *S. enterica* in T3), anti-biotin monoclonal antibody (for detection of *E. coli O157:H7* in T4), and anti-Cy5 monoclonal antibody (for detection of *L. monocytogenes* in T5). The distance between the control line and the test line was 3 mm. Each monoclonal antibody would be tuned to the optimal concentration. The anti-mouse polyclonal antibody (pAb) was immobilized on the control line (C-line) with 2.0 mg/mL. The assembled test strips were dried at 37 °C overnight. Finally, they were cut into 2.5 mm test strips with a cutter and stored in a vacuum bag at room temperature with desiccant.

### 2.5. Optimization of the RPA-LFIA Conditions

To improve the detection efficiency of RPA-LFIA, some key parameters were determined by optimization, including the primers concentration, incubation temperature, reaction time, and magnesium ion concentration. Five pairs of primer concentrations were verified in gradients of 150, 200, 250, 300, 350, and 400 nM to determine the optimal primers ratio. The reaction time and incubation temperature could also directly affect the results during RPA amplification. There were eight temperature gradients of 33, 34, 35, 36, 37, 38, 39, 40 °C that were investigated, and ten reaction time gradients of 2.5, 5, 7.5, 10, 12.5, 15, 17.5, 20, 22.5, and 25 min were compared to obtain the optimal reaction conditions. Moreover, seven different magnesium ion concentrations 0, 2.8, 5.6, 8.4, 11.2, 14, and 16.8 nM, were added to the mixture to reduce the activation energy for a faster priming reaction. Finally, the concentration of the fixed antibodies could also affect the efficiency of the immunoassay. The optimal concentration of each antibody was obtained by comparing various combinations (0.2–1.5 mg/mL) based on the intensity of the T-line on the dipstick.

### 2.6. Specificity and Sensitivity of Multiple RPA-LFIA Assay

The genomic DNA extracted from the 48 common foodborne strains (10^7^ CFU/mL) listed in Table 1 was selected to test the specificity. The presence or absence of cross-reaction for the multiple-RPA-LFIA method was observed by testing single pathogenic DNA, such as that from *V. parahaemolyticus*, *S. aureus*, *S. enterica*, *E. coli O157:H7* and *L. monocytogenes* and other foodborne strains.

In the sensitivity of the RPA-LFIA analysis, the five foodborne strains in the mid-exponential growth phase were tenfold serially diluted from 10^7^ to 10^0^ CFU/mL, and the genomic DNA was extracted for multiple-RPA-LFIA analysis. The five reference foodborne strains at the same concentration level were mixed in equal volume. All experiments were triplicated independently, and each strip was scanned three times.

### 2.7. Evaluation of Multiple RPA-LFIA in Artificially Contaminated Food Samples

The five reference foodborne strains were cultured in their respective fluid media at 37 °C overnight. The cultures were used to prepare the reference strains of various concentrations. Samples of chicken, pork, beef, milk, shrimp, and fish were purchased from local markets (Hangzhou, China). These samples were certified as negative for five reference foodborne strains according to the bacteriological analytical manual (BAM, Chapter 5, formulated by the Food and Drug Administration) [29]. Each type of meat (25.0 g ± 0.1 g) was homogenized in 225 mL of 4 M NaCl at 8000 rpm for 1 min under sterile conditions. For milk, a 25 mL volume was mixed with 225 mL of buffered peptone water (BPW) (10 g/L peptone, 5 g/L sodium chloride, 9 g/L disodium hydrogen phosphate dodecahydrate, 1.5 g/L potassium dihydrogen phosphate, pH 7.2). After mixing evenly, each homogenate was separately spiked with 10^4^, 10^3^, 10^2^, and 10^1^ CFU/mL of five reference foodborne strains. Subsequently, the genomic DNA from each spiked sample was determined using automatic extraction and purification by collecting, transferring, and releasing magnetic bead steps with a mini automatic nucleic acid extractor (for method refer to Section 2.1).

### 2.8. Field Samples Testing

Eighty food samples were randomly purchased from the local market, including chicken, raw pork, eggs, milk, cheese, raw shrimp, fish, codfish, broccoli, and fruit juice. After each sample was weighed (25.0 g ± 0.1 g or 25 mL ± 0.1 mL), 225 mL BPW buffer was added and fully mixed by rotation. Then, after complete homogenization, the mixture was concentrated at 37 °C and oscillated for 200 rpm for 16 h. After enrichment, 1 mL supernatant was extracted as described in Section 2.1. All samples were detected using multiple RPA-LFIA and BAM assays.

### 2.9. Statistical Analysis

Data collected from the RPA-LFIA assay were scanned using the test strip reader and analyzed with the TSR-200 reader software (Allsheng Instruments Co. Ltd., Hangzhou, China) and Microsoft Excel software (Microsoft Inc., Washington, DC, USA). The photoelectric sensor was used to measure the intensity of reflected light, and it could convert the light signals from the test (T) line and the control (C) line into electrical signals with the “T value” and “C value”. The band intensity was expressed in terms of the ratio of the T value to the C value (T/C value), and the standard curve was drawn according to the logarithm of the bacterial culture concentration and the T/C value. The recovery rate was calculated using the average based on the following equation:$$\mathrm{Recovery}\mathrm{rate}=\frac{\mathrm{Detection}\mathrm{concentration}}{\mathrm{Spiked}\mathrm{concentration}}\times 100\%$$

## 3. Results

### 3.1. Assay Principle

In this study, a mini automatic nucleic acid extractor was used efficiently and conveniently to extract nucleic acid from food samples. The instrument used modified magnetic beads to isolate and purify genomic DNA (Figure 1a). The extraction and purification work was carried out by the transfer and elution of magnetic beads.

To distinguish different amplicons, five sets of specific primers were designed by labeling different chemical groups. Subsequently, the target genes of the five reference foodborne pathogens were amplified using the TwistDx basic kit (TwistDx, UK). The amplification products were displayed with the LFIA assay. On the test strip, the C-line fixed the anti-digoxin antibody, and the T-line (T1–T5) fixed the anti-TET antibody, anti-FAM antibody, anti-TAMRA antibody, anti-biotin antibody, and anti-Cy5 antibody in the NC membrane. As shown in Figure 1b, after the amplification, products were added to the sample pad, and the capillary force made them move to the NC membrane. The labeled products bound to the corresponding antibody on the detection line to produce a red line. The uncaptured AuNPs were passed through and immobilized by antibodies on the C-line.

The quantitative detection was performed with the help of a test strip reader. The intensity of the signal from the T-line and C-line was obtained by transforming the optical signal to the electrical signal, and the standard curve was plotted according to the T/C value and logarithm of the concentration of bacterial cultures.

### 3.2. Establishment and Optimization of Multiple RPA-LFIA Assay

To achieve optimal amplification, critical parameters were used. The entire experiment was repeated three times. After the preliminary experiment, it was found that 25 min incubation was sufficient to observe bands clearly. In the beginning, the concentration of the five foodborne pathogens’ primers was selected at 150 nM. The bands were faintly discernible, as shown in Figure 2a. Hence, the concentration of primers was successively increased (from 150 to 400 nM) and assessed. Finally, the concentration of primers for *V. parahaemolyticus* was 250 nM, for *S. aureus* it was 350 nM, for *S. enterica* it was 400 nM, for *E. coli O157:H7* it was 250 nM, and for *L. monocytogenes* it was 400 nM. 

Secondly, eight temperature gradients were subsequently set to determine the optimal incubation temperature. The results (Figure 2b) showed that the brightest T-lines were obtained in the range of 37–39 °C. When the temperature exceeded 39 °C, the bands gradually weakened. Therefore, 37 °C was selected as the optimal reaction temperature.

Next, the effect of time on the amplification efficiency was validated. As shown in Figure 2c, different reaction times of 2.5, 5, 7.5, 10, 12.5, 15, 17.5, 20, 22.5 and 25 min were tested. The results showed that all significant T-lines were visible after 12.5 min. Through the strip reader, signals were stable after 15 min. After that, the reaction time (15 min) and incubation temperature (37 °C) were unified in the subsequent optimization experiments.

Seven different magnesium ion concentrations of 0, 2.8, 5.6, 8.4, 11.2, 14 and 16.8 mM were then tested. All five T-lines were brighter and more stable when the concentrations ranged from 14 to 16.8 mM, as shown in Figure 2d. A value of 14 mM was chosen as the optimal concentration of the magnesium ions.

### 3.3. Sensitivity and Specificity of the Multiple RPA-LFIA Assay

The tenfold serial dilutions of pure bacterial solutions were prepared to evaluate the sensitivity of multiple RPA-LFIA assays. An equal volume of bacterial solutions in the same concentration level were mixed. The genomic DNA was extracted from the mixture (for method refer to Section 2.1). The tests were repeated three times and they were evaluated under optimal conditions. In Figure 3a, the brightness of bands of five foodborne pathogens decreased gradually with the dilution. There was no clear T-line when the concentrations below were 10^1^ CFU/mL. The visual detection limits of RPA-LFIA for the five foodborne pathogens were 2.4 × 10^1^ CFU/mL (for *V. parahaemolyticus*), 7.1 × 10^1^ CFU/mL (for *S. aureus*), 4.5 × 10^1^ CFU/mL (for *S. enterica*), 5.1 × 10^1^ CFU/mL (for *E. coli O157:H7*), and 2.7 × 10^1^ CFU/mL (for *L. monocytogenes*). The average sensitivity of the five strains was 4.36 × 10^1^ CFU/mL. The T/C value from the signal intensity was obtained quantitatively through the strip reader. The standard linear curves (Figure 3b) have correlation coefficients of determination for *V. parahaemolyticus* (R^2^ = 0.9811), *S. aureus* (R^2^ = 0.9865), *S. enterica* (R^2^ = 0.9815), *E. coli O157:H7* (R^2^ = 0.9876), and *L. monocytogenes* (R^2^ = 0.9891).

There were 48 common foodborne strains (Table 1) that were selected to evaluate the specificity of the multiple RPA-LFIA assay, including six strains of *V. parahaemolyticus*, six strains of *S. aureus*, seven strains of *S. enterica*, four strains of *E. coli O157:H7*, six strains of *L. monocytogenes*, and nineteen other foodborne strains. The DNA from 48 pathogenic bacteria was extracted and added to the RPA reaction to determine the specificity of the multiple RPA-LFIA method. The results (Figure 4a) showed that only 29 targeted foodborne pathogens were positive in testing, and the others were negative. Figure 4b shows that the pathogenic bacteria alone have a clear visual band in the specific detection of the test strips. This indicated that there was no cross-reactivity among the five target genes.

### 3.4. Application of RPA-LFIA in Spiked Samples

A total of six spiked samples were identified using the RPA-LFIA technique. Chicken, pork, beef, milk, shrimp, and fish were prepared in advance as artificially contaminated food samples. All samples were contaminated by adding different concentrations of *V. parahaemolyticus*, *S. aureus*, *S. enterica*, *E. coli O157:H7*, and *L. monocytogenes.* After inoculation, samples were tested for BAM. The recovery rates in spiked samples were 92.0–102.3% (for *V. parahaemolyticus*), 90.5–104.5% (for *S. aureus*), 90.6–104.0% (for *S. enterica*), 96.9–101.0% (for *E. coli O157:H7*), and 91.6–104.4% (for *L. monocytogenes*), as shown in Appendix A. The inoculated samples were tested by BAM and the results were consistent with the RPA-LFIA conclusion.

### 3.5. Detection of Field Samples

Eighty different samples were analyzed using multiple RPA-LFIA and BAM methods. The positive detection rate of the RPA-LFIA assay for *V. parahaemolyticus* was 3.75%, for *S. enterica* it was 2.5%, for *E. coli O157:H7* it was 1.25%, and *S. aureus* and *L. monocytogenes* were both 0%, respectively. The results (shown in Table 3) demonstrated perfect consistency between the two methods.

## 4. Discussion

With the development and diversification of food supply chains in recent years, diseases caused by foodborne pathogens have been a prominent problem for food safety. A variety of toxicants are produced in the process of food pollution which can harm human health. The *V. parahaemolyticus*, *S. aureus*, *S. enterica*, *E. coli O157:H7*, and *L. monocytogenes* are five common foodborne pathogens that can cause various foodborne diseases. In fact, the fresh food came from a wide range of markets, and the complicated food processing made it easily become simultaneously contaminated with multiple pathogens. Therefore, rapid detection and identification of the existence of various pathogens in food is an important reason for timely and efficient control of food poisoning so as to prevent the spread of pathogens.

Traditional culture detection technologies require a large amount of time to implement, and most of them use a complicated process [30]. Molecular diagnostic technology can greatly shorten the testing period. Among them, PCR requires the thermal cycler to accurately control temperature, but it is not suitable for application in the field [31]. Isothermal amplification technologies have been developed in recent years, including LAMP, nucleic acid sequence-based amplification (NASBA), and RPA. These methods overcame the dependence on the appointed instrument and work at constant temperatures [32]. The lower tolerance for samples of NASBA and the difficulty for multiples of LAMP have limited their widespread application. The RPA assay, first reported in 2006, has both the advantages mentioned above and it was more suitable for use in low-resource areas at comparatively lower temperatures (37–42 °C) and within shorter working times [33]. Due to the potential for cross contamination, it has received great interest due to its ability to simultaneously amplify and detect multiple targets, especially for purposes of monitoring and diagnosis [32]. The reported multiplex RPA has been successfully applied in several fields, including animal epidemic disease [34], plant pathogens [35], and pathogenic bacterium [26]. However, there are few reports reporting on the simultaneous testing of five food-borne pathogens.

In this study, *V. parahaemolyticus*, *S. aureus*, *S. enterica*, *E. coli O157:H7*, and *L. monocytogenes* were selected as reference pathogens based on the number and effects of foodborne disease outbreaks. The Auto-Pure Mini instrument was used to improve efficiency and obtain extraction stability. Using the means of the specific adsorption function of the modified magnetic beads, the automatic extraction could increase throughputs and maintain purity from food samples. Compared with previous manual methods (shown in Appendix A), it was noteworthy that the automatic extractor overcame the limitations in terms of stability, simplicity, and rapidity. For the convenience of on-site testing, the lateral flow test strips were combined with the RPA assay. The amplification products were visually read via AuNPs color signals, and they could be quantitatively analyzed using a strip reader. Combined with multi-dimensional spraying technology in LFIA, it could realize multiple detection capabilities. Meanwhile, the key parameters were evaluated and optimized [36]. The competitive effect of different primer pairs on recombinant proteins caused the inconsistent amplification efficiency. Therefore, the concentration ratio of the primers for different target genomic DNA needed to be adjusted appropriately. The optimal five primer concentrations for *V. parahaemolyticus*, *S. aureus*, *S. enterica*, *E. coli O157:H7*, and *L. monocytogenes* were finally determined as 250, 350, 400, 250, and 400 nM. In addition, the optimal reaction temperature and time were determined at 37 °C for 15 min. Beyond that, cross-reaction and inhibition increased with the number of T-lines during the experiment. Thus, the concentrations of labeled antibodies could also affect the immunoassay. The concentration of each fixed antibody was determined using a separate reaction, and it was properly increased based on the brightness of the individual T-lines and C-line. The different amounts were subsequently integrated into one strip. The final determined concentrations were 2.0 mg/mL of anti-mouse polyclonal antibody (pAb), 0.5 mg/mL of anti-TET monoclonal antibody, 0.2 mg/mL of anti-FMA monoclonal antibody, 0.5 mg/mL of anti-TAMRA monoclonal antibody, 0.4 mg/mL of anti-biotin monoclonal antibody, and 0.45 mg/mL of anti-Cy5 monoclonal antibody (Appendix A). The results of RPA-LFIA were quantified using the test strip reader and showed that the limit of detection (LOD) was 10^1^ CFU/mL. The current sensitivity and specificity assessments represent proof-of-concept work that should be pursued further in future studies with a larger number of strains. Compared with the other methods listed in Appendix A, the application of the test strip reader achieved the objective of quantitative detection, reduced artificial deviation and improved diagnostic accuracy. Furthermore, the recoveries of spiked samples were 90.5–104.5%. In brief, the RPA-LFIA presented in this paper was a promising method for detecting five foodborne pathogens. The established assay addresses the need for multiple detection combined with a pretreatment instrument to isolate genomic DNA, making it suitable for quantitative testing in the field, especially in resource-limited areas.

## 5. Conclusions

In this study, we described the combination of a mini automatic nucleic acid extractor and RPA-LFIA assay. It was not only able to stably extract the genomic DNA from the samples but also able to simultaneously detect *V. parahaemolyticus, S. aureus,*
*S. enterica, E. coli O157:H7*, and *L. monocytogenes* in food. The ability to detect multiple targets in the same reaction was successfully demonstrated by the efficient amplification of five foodborne pathogens. This demonstrates that the RPA-LFIA shows the best performance at 37 °C within 15 min. Without the requirement for the thermocycling process and the additional costly equipment, the assay simplified operation, which made it quite easy to master. The RPA-LFIA results could be directly visible, and it was also presented quantitatively by the strip reader. In general, the RPA-LFIA is a practical, simple-to-conduct, and easy-to-read method for the rapid detection of five foodborne pathogens in the field.

## Figures and Tables

**Figure 1 microorganisms-10-01352-f001:**
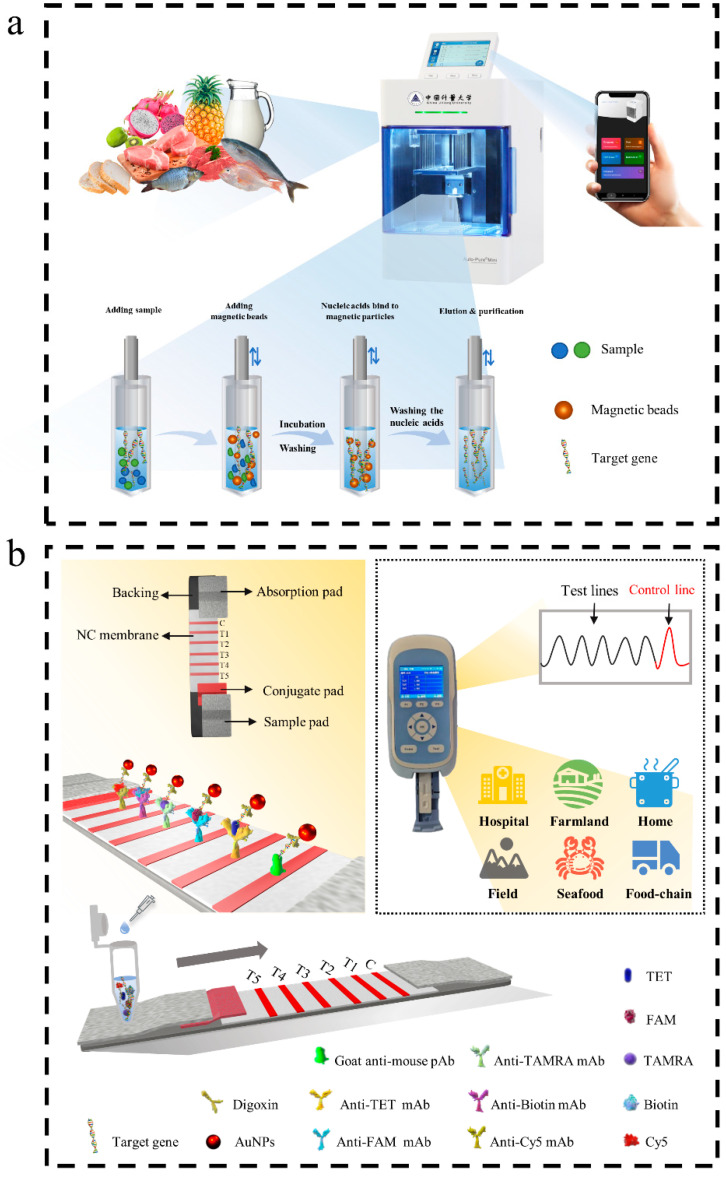
Schematic illustration of the RPA-LFIA for simultaneous detection of five targets. (**a**) Operation procedure of mini automatic nucleic acid extractor. (**b**) The working principle of the multiple RPA-LFIA assay. The meaning of Chinese characters is China Jiliang University.

**Figure 2 microorganisms-10-01352-f002:**
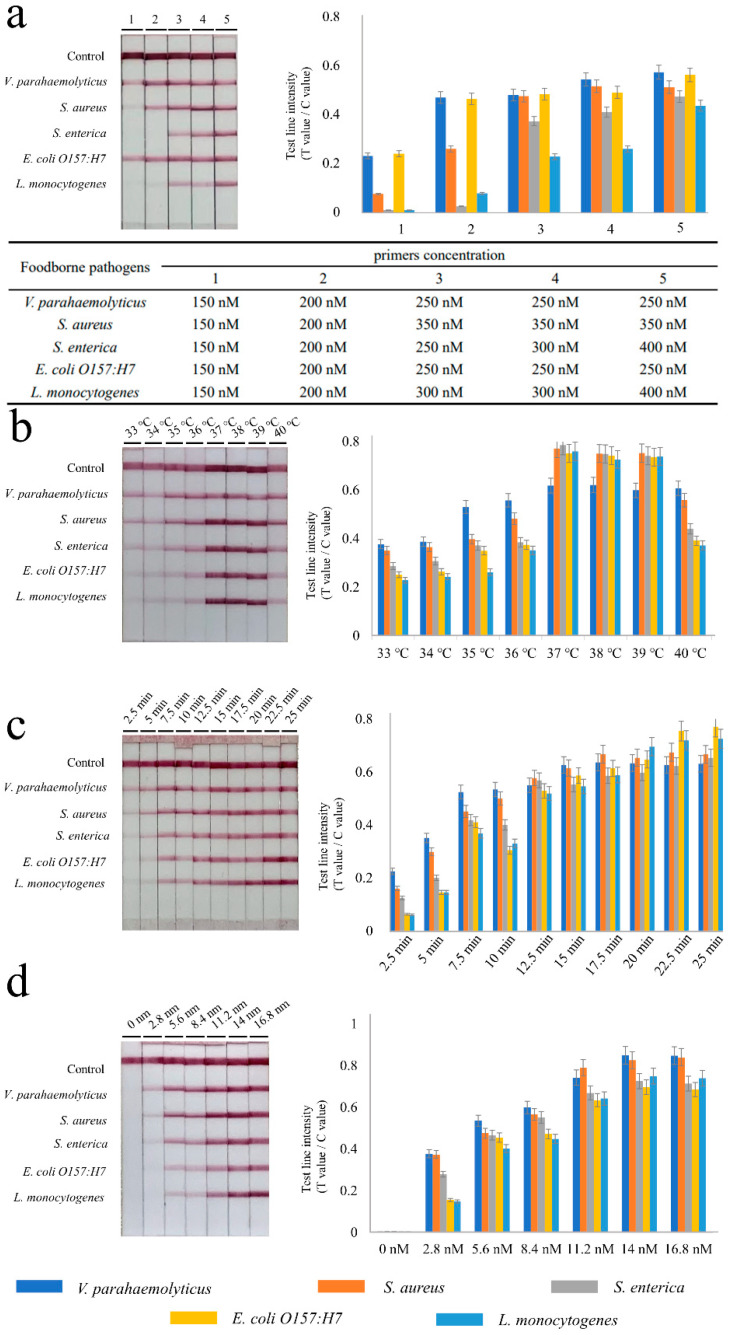
Optimization of recombinase polymerase amplification reactions. (**a**) Different primer concentrations and the corresponding primer concentrations are shown in the three-line table. (**b**) The results for the incubation temperature, (**c**) the results for the RPA reaction time, (**d**) the results for the magnesium ions concentration. The RPA products were detected using lateral flow immunoassay (LFIA) based on labeled colloidal gold (presented in left). The intensity of reflected light signal from the T-line and C-line on LFIA was obtained by the test strip reader. The test line intensity is defined as the ratio of the T value and the C value. The bar graph is based on the radio of different T-lines and C-line (presented on the right). Each parameter is shown on the top of the strips or in the bottom of the bar graph. Three replicates are shown in the experiment.

**Figure 3 microorganisms-10-01352-f003:**
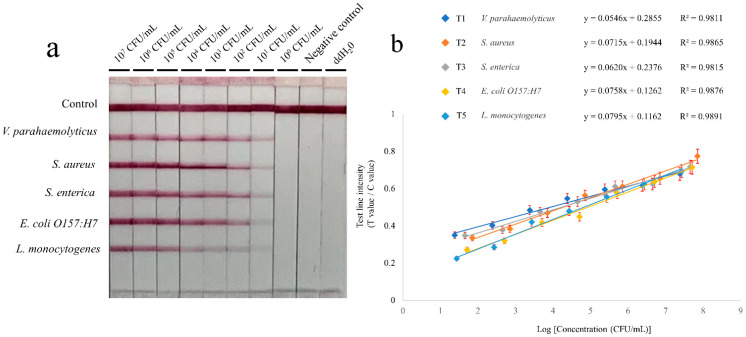
Reaction sensitivity of the RPA-LFIA. (**a**) Amplification results. The initial concentration of *V. parahaemolyticus*, *S. aureus*, *S. enterica*, *E. coli O157:H7*, and *L. monocytogenes* were 2.4 × 10^7^ CFU/mL, 7.1 × 10^7^ CFU/mL, 4.5 × 10^7^ CFU/mL, 5.1 × 10^7^ CFU/mL, and 2.7 × 10^7^ CFU/mL, respectively. After tenfold serial dilution, an equal volume of bacterial solutions in the same concentration level were mixed. The tests were repeated three times. (**b**) Standard curves. The sensitivity was evaluated with the logarithmic (log10) values ranging from 10^7^ to 10^0^ CFU/mL.

**Figure 4 microorganisms-10-01352-f004:**
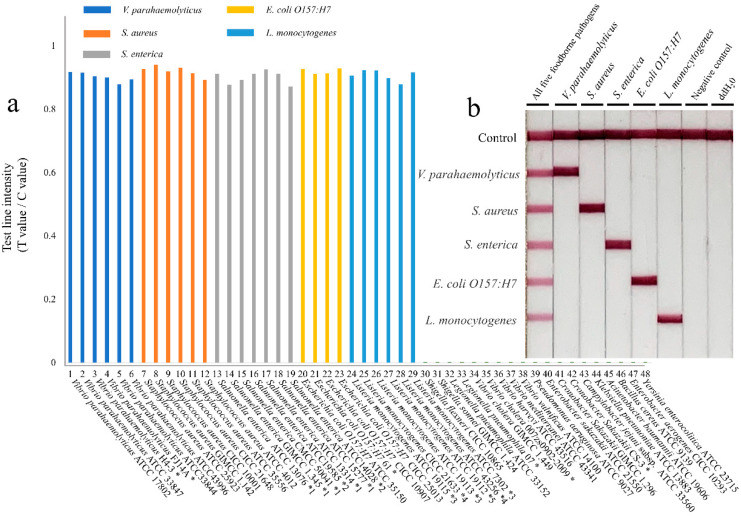
Specificity testing of the RPA-LFIA assay. (**a**) Results of the specific evaluation of the 48 foodborne strains. (**b**) Specific test results on the lateral flow dipstick. “*”: Presented by Zhoushan Entry-Exit Inspection and Quarantine Bureau, without serotype identification. “*1”: Serovar Enteritidis; “*2”: Serovar Typhimurium; “*3”: Serotype 4b; “*4”: Serotype 1/2a; “*5”: Serotype 1/2c.

**Table 1 microorganisms-10-01352-t001:** Information of bacterial strains used for specificity tests.

Species	ID of Strains	Multiple RPA-LFIA Test Results				
		*toxR*	*nuc*	*fimY*	*rfbE*	*hlyA*
*Vibrio parahaemolyticus*	ATCC 17802	+	-	-	-	-
*Vibrio parahaemolyticus*	ATCC 33847	+	-	-	-	-
*Vibrio parahaemolyticus*	H4-3 *	+	-	-	-	-
*Vibrio parahaemolyticus*	FJ14A *	+	-	-	-	-
*Vibrio parahaemolyticus*	ATCC33844	+	-	-	-	-
*Vibrio parahaemolyticus*	ATCC43996	+	-	-	-	-
*Staphylococcus aureus*	ATCC 25923	-	+	-	-	-
*Staphylococcus aureus*	GIMCC 1.142	-	+	-	-	-
*Staphylococcus aureus*	CICC 10001	-	+	-	-	-
*Staphylococcus aureus*	CICC 21648	-	+	-	-	-
*Staphylococcus aureus*	ATCC 35556	-	+	-	-	-
*Staphylococcus aureus*	ATCC 4012	-	+	-	-	-
*Salmonella enterica*	ATCC 13076 *1	-	-	+	-	-
*Salmonella enterica*	GIMCC 1.345 *1	-	-	+	-	-
*Salmonella enterica*	CMCC 50041 *1	-	-	+	-	-
*Salmonella enterica*	ATCC19585 *2	-	-	+	-	-
*Salmonella enterica*	ATCC 13314 *1	-	-	+	-	-
*Salmonella enterica*	ATCC 15277 *1	-	-	+	-	-
*Salmonella enterica*	ATCC14028 *2	-	-	+	-	-
*Escherichia coli O157:H7*	ATCC 35150	-	-	-	+	-
*Escherichia coli O157:H7*	61 *	-	-	-	+	-
*Escherichia coli O157:H7*	CICC 10907	-	-	-	+	-
*Escherichia coli O157:H7*	CICC 25013	-	-	-	+	-
*Listeria monocytogenes*	ATCC 19115 *3	-	-	-	-	+
*Listeria monocytogenes*	CICC 21633 *4	-	-	-	-	+
*Listeria monocytogenes*	ATCC 19113 *3	-	-	-	-	+
*Listeria monocytogenes*	ATCC 19112 *5	-	-	-	-	+
*Listeria monocytogenes*	ATCC 43256 *4	-	-	-	-	+
*Listeria monocytogenes*	ATCC 7302 *3	-	-	-	-	+
*Shigella flexneri*	CICC 10865	-	-	-	-	-
*Shigella sonnei*	GIMCC 1.424	-	-	-	-	-
*Legionella pneumophila*	ATCC 33152	-	-	-	-	-
*Legionella pneumophila*	07 *	-	-	-	-	-
*Vibrio cholera*	GIMCC 1.449	-	-	-	-	-
*Vibrio cholera*	007zs0902-2009 *	-	-	-	-	-
*Vibrio harveyi*	ATCC 43516	-	-	-	-	-
*Vibrio mediterranei*	ATCC 43341	-	-	-	-	-
*Vibrio vulnificus*	ATCC 14100	-	-	-	-	-
*Pseudomonas aeruginosa*	ATCC 9027	-	-	-	-	-
*Enterobacter sakazakii*	ATCC 21550	-	-	-	-	-
*Cronobacter Sakazakii*	GIMCC 1.296	-	-	-	-	-
*Cronobacter Sakazakii*	CS-3 *	-	-	-	-	-
*Campylobacter jejuni subsp.*	ATCC 33560	-	-	-	-	-
*Klebsiella pneumoniae*	ATCC 13883	-	-	-	-	-
*Acinetobacter baumannii*	ATCC 19606	-	-	-	-	-
*Bacillus cereus*	ATCC 9139	-	-	-	-	-
*Enterobacter aerogenes*	CICC 10293	-	-	-	-	-
*Yersinia enterocolitica*	ATCC 23715	-	-	-	-	-

ID: identifier; RPA: recombinase polymerase amplification; LFIA: lateral flow immunoassay; *toxR*: *toxR* gene (Genebank accession: GQ228073.1) of *Vibrio parahaemolyticus*; *nuc*: *nuc* gene (Genebank accession: EF529607.1) of *Staphylococcus aureus*; *fimY: fimY* gene (Genebank accession: JQ665438.1) of *Salmonella enterica*; *rfbE: rfbE* gene (Genebank accession: AE005429) of *Escherichia coli O157:H7*; *hlyA: hlyA* gene (Genebank accession: HM58959) of *Listeria monocytogenes*., GIMCC: Guangdong Microbiology Culture Center, Guangdong, China; ATCC: American Type Culture Collection, Virginia, USA; CICC: China Center of Industrial Culture Collection, Shanghai, China; “*”: Presented by Zhoushan Entry-Exit Inspection and Quarantine Bureau, without serotype identification. “*1”: Serovar Enteritidis; “*2”: Serovar Typhimurium; “*3”: Serotype 4b; “*4”: Serotype 1/2a; “*5”: Serotype 1/2c. “+”: positive result; “-”: negative result.

**Table 2 microorganisms-10-01352-t002:** Sequences of five foodborne pathogens for RPA primers.

Target Name	Target Name	Sequence (5′-3′)	Modifications	Amplification Size	Reference
*V. parahaemolyticus*	*toxR*-RPA F(forward primer)	TTTGTTTGGCGTGAGCAAGGTTTTGAGGTG	5′-TET	230 bp	[26]
	*toxR*-RPA R(reverse primer)	GCAGAGGCGTCATTGTTATCAGAAGCAGGT	5′-Digoxin		
*S. aureus*	*nuc*-RPA F(forward primer)	CTTATAGGGATGGCTATCAGTAATGTTTCG	5′-FAM	158 bp	
	*nuc*-RPA R(reverse primer)	CCACTTCTATTTACGCCGTTATCTGTTTGT	5′-Digoxin		
*S. enterica*	*fimY*-RPA F(forward primer)	TATCAGATAAAACCTCCGCTATAACACAGT	5′-TAMRA	133 bp	
	*fimY*-RPA R(reverse primer)	CTTTCCGATAAGCGAGGTTTGGAGGCTGAT	5′-Digoxin		
*E. coli O157:H7*	*rfbE*-RPA F(forward primer)	TATCTGCAAGGTGATTCCTTGATGGTCTCA	5′-Biotin	176 bp	[27]
	*rfbE*-RPA R(reverse primer)	AGGCCAGTTACCATCCTCAGCTATAGGGTG	5′-Digoxin		
*L. monocytogenes*	*hlyA*-RPA F(forward primer)	CGATCACTCTGGAGGATACGTTGCTCAATT	5′-Cy5	154 bp	
	*hlyA*-RPA R(reverse primer)	TTACCAGGCAAATAGATGGACGATGTGAAA	5′-Digoxin		

**Table 3 microorganisms-10-01352-t003:** Comparison of field samples detected by the RPA-LFIA and BAM methods.

Samples	No. of Samples	*V. parahaemolyticus*		*S. aureus*		*S. enterica*		*E. coli O157:H7*		*L. monocytogenes*	
		RPA-LFIA	BAM	RPA-LFIA	BAM	RPA-LFIA	BAM	RPA-LFIA	BAM	RPA-LFIA	BAM
Milk	12	-	-	-	-	-	-	-	-	-	-
Raw pork	10	-	-	-	-	-	-	1	1	-	-
Eggs	9	-	-	-	-	-	-	-	-	-	-
Chicken	9	-	-	-	-	-	-	-	-	-	-
Cheese	9	-	-	-	-	-	-	-	-	-	-
Raw shrimp	8	3	3	-	-	2	2	-	-	-	-
Fish	7	-	-	-	-	-	-	-	-	-	-
Codfish	6	-	-	-	-	-	-	-	-	-	-
Broccoli	5	-	-	-	-	-	-	-	-	-	-
Fruit juice	5	-	-	-	-	-	-	-	-	-	-
Total	80	3	3	-	-	2	2	1	1	-	-
Positive Detection rate	/	3.75%	3.75%	0%	0%	2.50%	2.50%	1.25%	1.25%	0%	0%

“-”: negative result.

## Data Availability

The original contributions presented in the study are included in the article/Appendix A, further inquiries can be directed to the corresponding author.

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
