# Peer review of "Simultaneous Detection of Five Foodborne Pathogens Using a Mini Automatic Nucleic Acid Extractor Combined with Recombinase Polymerase Amplification and Lateral Flow Immunoassay"

_microorganisms, 2022, doi:10.3390/microorganisms10071352_

Round 1

Reviewer 1 Report

Enteric pathogens are a major cause of foodborne disease outbreaks. An accurate and rapid detection  system is required for identification of foodborne pathogens. The authors propose to use an automatic extraction of DNA from food samples. Specific genes are amplified by RPA (recombinant polymerase amplification). The authors used specific primers for nuc (Staphylococcus aureus),  toxR (Vibrio parahaemolyticus), rfbE (Escherichia coli O157:H7), hlyA (Listeria monocytogenes), and fimY (Salmonella Enteritidis) genes in the DNA amplification. The primer ratios, incubation temperature, reaction time, the concentration of magnesium ions in buffer were optimized.

C 1. Although the results support the conclusions of the authors. The data is poorly presented in fig. 2. The legend does not described the figure in detail. The titles in the figure are tiny and contains a lot of information. Please use the legend of the figure to describe the strains, etc.  

To detect the RPA products, the authors propose to use lateral flow immunoassay (LFIA). They use labeled colloidal gold. The particles are incubated with anti-TET (for Vibrio parahaemolyticus), anti-FAM (for Staphylococcus aureus), anti-TAMRA (for Salmonella Enteritidis), anti-biotin (for Escherichia coli O157:H7) and anti-Cy5 (for Listeria monocytogenes). The RPA products were added to the sample pad, the capillary force made them move to the NC membrane.

C2. The preparation of LFIA membranes is poorly described as well as the treatment after the incubation with the RPA products to avoid background. Please describe this part in detail in the manuscript.

It is remarkable that 15 min of incubation at 37oC is sufficient to observe bands a 37oC. These findings support the authors conclusion that detection of the pathogens does not required additional equipment (only a strip reader) and it is rapid. The detection limits for the RPA-LFIA is  less of 100 CFU (showed in Fig. 4).

C3. The figure 4 clearly showed that there is no cross-reactivity among genes. The graph in figure 4 does not have labeling and it is not described in the manuscript.

To validate the RPA-LFIA, the authors analyzed eighty different samples with RPA-LFIA and BAM methods. The results were comparable between the two methods.

C4. Since the success of a procedure depends on detection limits, can  the authors compere both procedures with low bacteria load (10 and 100 CFU)?. 

Author Response

Dear reviewer:

Thanks for your comments. We have added all the necessary information to clarify the findings as well as the details of the graph. We updated figure 2, figure 3, figure 4, figure S1, and table 2. At the same time, we renumbered table 3, table S1, table S2 and table S3. All details can be seen in the revised manuscript with red highlights. We have improved the quality of our article language by professional English editing service of MDPI (english-43677). All details can be seen in the revised manuscript with blue highlights.

Author Response

(The authors gave the same response as above.)

Reviewer 3 Report

The manuscript describes original research on a new method to simultaneously detect quantitively five major human foodborne pathogens in food, based on combination of RFA-LFIA. Some clarifications and corrections are needed:

* The document must be thoroughly revised to edited the species name: it should be written out in full when it first appears in Introduction and in subsequently text the abbreviated form should be used. Examples: L42,43, 46, 49, 52, 116-118, 153-158, 186-189, 212-213, 310-316,, 322-324, 333-338.

* Introduction section could be shortened and should be improved, and English edited. Specifically the past tense is used inappropriately in numerous sentences.

*L 35   Reference [2] doesn’t seems to be adequate to state worldwide deaths due to foodborne illnesses.

* Figure 2 is too small and its analysis its difficult, thus needs to be replaced/amplified

* Table 3 – inoculation levels and concentration detected for each pathogens in different food is presented in CFU/mL – shouldn’t be CFU/g as solid food samples were used (except for milk)? Please clarify.

Check typos:

L 42, 50, 66, 74, 181   Delete “0” at the end of the sentences

L238 check parentheses position

Author Response

(The authors gave the same response as above.)

Round 2

Reviewer 1 Report

The authors have responded to most of the criticism. In general terms, the figures were reorganized, the legends improved with more information, and the spellings were corrected. Please, can the authors include statistical analysis between the groups?

Author Response

Dear reviewer:

Dear reviewer:

Thank you very much for the time that you spent on reviewing our manuscript. Those comments are all valuable and helpful for revising and improving our paper. We have added all necessary information to clarify the findings and the details of figures and tables. We have edited the additional text and modified the formatting in the supplementary material. All details can be seen in the revised manuscript with red highlights. Please check the attachment.

Reviewer 2 Report

The inclusion of the supplemental tables has improved the manuscript, however there are still problems with table formatting. 

The additional text requires further editing.

No new experiments have been conducted to strengthen the impact and interest in this manuscript.

Author Response

Dear reviewer:

Thank you very much for the time that you spent on reviewing our manuscript. Those comments are all valuable and helpful for revising and improving our paper. We have added all necessary information to clarify the findings and the details of figures and tables. We have edited the additional text and modified the formatting in the supplementary material. All details can be seen in the revised manuscript with red highlights. Please check the attachment.
